# Optimal Method for Test and Repair Memories Using Redundancy Mechanism for SoC

**DOI:** 10.3390/mi12070811

**Published:** 2021-07-10

**Authors:** Suleman Alnatheer, Mohammed Altaf Ahmed

**Affiliations:** Department of Computer Engineering, College of Computer Engineering & Sciences, Prince Sattam Bin Abdulaziz University, Al-Kharj 11942, Saudi Arabia; s.alnatheer@psau.edu.sa

**Keywords:** SoC, BISR, BIRA, memory test and repair, repair-rate

## Abstract

The current system-on-chip (SoC)-based devices uses embedded memories of enormous size. Most of these systems’ area is dense with memories and promotes different types of faults appearance in memory. The memory faults become a severe issue when they affect the yield of the product. A memory-test and -repair scheme is an attractive solution to tackle this kind of problem. The built-in self-repair (BISR) scheme is a prominent method to handle this issue. The BISR scheme is widely used to repair the defective memories for an SoC-based system. It uses a built-in redundancy analysis (BIRA) circuit to allocate the redundancy when defects appear in the memory. The data are accessed from the redundancy allocation when the faulty memory is operative. Thus, this BIRA scheme affects the area overhead for the BISR circuit when it integrates to the SoC. The spare row and spare column–based BISR method is proposed to receive the optimal repair rate with a low area overhead. It tests the memories for almost all the fault types and repairs the memory by using spare rows and columns. The proposed BISR block’s performance was measured for the optimal repair rate and the area overhead. The area overhead, timing, and repair rate were compared with the other approaches. Furthermore, the study noticed that the repair rate and area overhead would increase by increasing the spare-row/column allocation.

## 1. Introduction

The recent SoC-based devices play a more important role as technology enhances day by day. These modern SoC designs are dense with memory, and the users need more promising features from their devices. A smooth-functioning memory-test algorithm and architecture are required to maintain the product’s reputation. Present system-on-chip (SoC) designs consist of embedded memory in a large portion. The embedded memory area in recent SoC-based devices is higher and is approximately equal to 95% of the total chip area [1,2,3]. Due to the high density of memory, there is a high probability of defects in SoC.

Furthermore, the memories are more prone to faults than the actual logic, as memories do not consist of the logic elements such as flip-flops [3,4], and the defects in the embedded memory of the devices or the systems can cause a critical error. Therefore, regress testing embedded memories in today’s complex SoC-based systems becomes necessary to retain the products’ reputation in the market. Thus, the SoC-based product yield is drastically affected by the memory in the chip. The effective yield-improvement method becomes essential for SoC design. Testing memories for faults and repairing defected memory methods play a vital role in improving the SoC design yield [5]. Memory-fault test and repair are popular techniques for yield improvement [6], and the built-in self-repair (BISR) is a widespread scheme to enhance the yield of the memory-based product.

Memory test and repair are two separate processes involved in yield enhancement for any modern SoC design semiconductor memories. Memory built-in self-test (MBIST) is a verified and reliable method for testing embedded memory [7,8,9,10], whereby memories are tested for fault and the fault types using sophisticated March algorithms. The MBIST controller usually works on test algorithms for finding defects and their types in embedded memories [11,12]. Testing the embedded memories by the test-pattern generator (TPG), using a scan chain method, is proposed in the research [13,14] to target less power consumption. Test time and test power are analyzed by the proposed scan chain architecture and LFSR. The LFSR is used to produce the test pattern by using a set of vectors for achieving the timing to test.

The failure information generated by the MBIST design during the testing process is available for the built-in self-repair (BISR) block after completing the test. The BISR circuitry uses this memory failure information to repair the defective memory cell. The BISR method is common and popular for memory yield improvement in SoC [15,16,17,18,19]. Commonly BISR method uses a built-in self-test (BIST) and the built-in redundancy analysis (BIRA) to test and repair the memory in any SoC design. Conventionally memory repair was carried out by two methods, memory tester and laser equipment [20]. Both of these methods are very expensive, time-consuming, and use extra hardware to repair the memory for fault, thus not in use. Therefore, the only BISR is the recommended solution for memory repair and thus yields improvement.

The multiple memories, test, and repair scheme is proposed targeting the low area and time [15,16]. Comprehensive real-time exhaustive search test and analysis (CRESTA) is the conventional algorithm offered to repair the embedded memory for faults [17]. It covers all the redundancy allocations using numerous sub-analyzers. In this way, it is better to achieve the optimum repair rate with no additional test time other than test procedures. On the other hand, the area overhead drastically increases due to various test redundancies and several sub-analyzers [17]. The repair rate and the area overhead both are equally important while dealing with the embedded memories. The repair rate improves the yield of the memory devices, where area overhead reduces the devices’ cost. The recent trend is high-density memories, and it requires an extensive redundancy analysis (RA) time with a large area overhead. Thus, the cost of the memory-based systems depends on the low area overhead with fast RA BIRA. Therefore, the tradeoff between BIRA (for repair rate) and low area overhead becomes the hot cake in the research field. It becomes necessary to maintain this characteristic to develop the product’s reputation in the market.

Ample research has been conducted for maintaining the low area and high repaired rate characteristics by the proposed methods of BISR, and it has achieved a suitable figure. However, still, it needs improvement, and it is possible to enhance this characteristic further. Therefore, a slight change in the technique to improve the test and repair method for the memory will help in increasing the product yield. This research tried to reduce the area overhead with a reasonable repaired rate and fault coverage by the proposed BIST and BISR. The contributions made by this research are listed below.

Implementation of BIST for critical testing by using the proposed March algorithm to improve a product’s quality. This memory BIST finds the faults and types in the memory to improve the fault coverage.A time- and cost-effective BISR solution to repair the faulty cells of the memory under test (MUT). The BIST controller generates the fault cells’ information to use spare memory. The BISR block performed the replacement of the faulty cells by redundant blocks, which are fault-free.

Thus, the BISR block allows defective cells to be replaced with the spare block without losing original data if memory contains critical information. Therefore, it ensures product reliability at a low cost.

## 2. Proposed Architecture

The proposed architecture of the built-in self-repair method is shown in Figure 1. The BISR block mainly consists of two phases, namely the memory-test phase and the fault-repair phase. The memory-test phase tests the memory for faults by the BIST method, and the fault-repair phase repairs the detected defects to provide error-free memory.

### 2.1. Fault Testing

The proposed memory BIST algorithm is used for fault testing. It diagnoses the memory for failures and preparing fail information for the repair block.

#### 2.1.1. Proposed Algorithm

To deliver high-grade SoC products, the SoC manufacturer needs high fault coverage. Therefore, most manufacturers try to implement a sophisticated memory BIST algorithm to reach reasonable percent fault coverage. Most memory BIST schemes are based on the marching algorithm to test the embedded memories and get significant fault coverage. Generally, a complex March algorithm can detect fault types, such as stuck-at fault, address decoder fault, transition faults, and some coupling faults. The current product’s increased chip density and technology result in new fault types in the SoC, such as stuck-open fault and neighborhood pattern sensitive faults (NPSFs). The designer needs to provide the algorithm carefully with these advancements in technology.

This research proposed a novel memory BIST algorithm to test the various memories for multiple faults in an SoC-based device. The memory BIST algorithm is based on March elements and is used to test the memories by applying different patterns, as mentioned in the algorithm. The algorithm itself writes various patterns, such as 0s and 1s, to the different memory locations of memory under test (MUT). As per the element written in parenthesis, each operation is performed at every memory location, one after another. The algorithm is marching 0s and 1s patterns in ascending or descending order of address sequence for the regress testing of the memory. A comparison with the original pattern (all 0s or all 1s) occurs during every read operation. While comparing if mismatches arise, the result will record into the failure information register. The March-sift operations continue after displaying the results from the next consecutive memory location, and it repeats the same until it reaches the last location. The proposed March-sift algorithm is expressed below, in Equation Box (1), and the operations are mentioned in Table 1.
{ sift0: ↕ (w_0_); sift1: ↑ (r_0_, w_1_); sift2: ↓ (r_1_, w_0_, r_0_); sift3: ↑ (r_0_, w_1_);sift4:↑ (r_1_, w_0_); sift5: ↓ (r_0_, w_0_, r_0_); sift6: ↑ (r_0_, w_1_, r_1_); sift7: ↕ (r_1_);}(1)

#### 2.1.2. Memory BIST Controller

Memory BIST controller works on the patterns of a March-sift algorithm. The March-sift algorithm consists of eight elements to perform read–write operations on the selected memory under test (MUT). Each element of the algorithm is considered a separate state. The controller state machine is designed by considering eight states and two extra states to start the test operation and store fails memory status. The memory BIST controller state machine is shown in Figure 2. The elements of the algorithm are as follows.

sift0: In the first step of test conduction, the writing operation performs at all the MUT locations. The address sequence is not essential in this state. It can be started in either direction, ascending or descending order.

sift1: In this step, two operations performed one after another at each memory location under test. Before coming to this state, memory is full of zeros. The first read zero will perform and compared with the desired zero patterns. The write ones will follow after reading zero at each location in ascending order. During every read, a comparison takes place with the original data pattern. While comparing if a mismatch takes place, the result will display in the fail information state, and the state machine jumps to the next memory location and repeat the same operations. During the comparison pass, the state machine directly jumps to the next address location.

sift2: At this step, three operations (read 1s, write 0s, and read 0s) are performed sequentially at each memory location (in descending order). Starting with the highest address, the first operation (r1) reads the memory location, where the expected value is one, and compares the reading value with the desired one pattern. If a mismatch occurs, the result will be stored in the status register. The second operation (w0) writes or fills the memory location with zero, followed by (r0). After completing the functions of sift2, the state machine repeats the same actions at the next address location and all remaining addresses until it reaches the lowest address of the memory locations under test.

sift3: At this step, the two operations are sequentially performed throughout the memory in ascending address order. Starting with the zero address, the first operation (r0) followed by (w1) write 1s. During a read operation of a memory location, the reading data value will compare with the desired zero patterns. If a mismatch of any bit occurs, the state machine will record the result. After these operations, the state machine jumps to the next consecutive location and repeats the same procedures until it reaches the memory’s highest address location.

sift4: At this step, the two operations performed sequentially throughout all memory locations under test (in descending order). Starting with the highest address, the first operation (r1) reads the memory location, where the expected value is one, and compares the reading value with the desired one pattern. If a mismatch occurs, the result will be recorded. The second operation (w0) writes or fills the memory location with zero. After these operations, the state machine jumps to the next memory location and repeats the same activities until it reaches the lowest address location of the memory under test.

sift5: At this step, the operation is performed throughout all memory locations under test (in descending address ordering). Starting by the lowest address, the action (r0) reads the memory location, then writes 0s (w0), and reads 0s (r0) at the same location are performed. The reading value will be compared with the original zero patterns and recorded when a mismatch occurs every time. After reading and analyzing, the state machine jumps to the next memory location, repeats the same operations, and continues until it reaches the highest memory locations.

sift6: Similarly, as the state sift5, all operations performed with a change in the elements w0 to w1 and r0 to r1.

sift7: At this step, the operation performed throughout all memory locations under test (in ascending or descending address ordering) direction is not essential. Starting by the lowest address, the action (r1) reads the memory location, where the expected value is one. The reading value will be compared with the original 1s pattern and recorded when mismatch.

Memory BIST operations start when the state machine gets the start signal to perform March-sift functions. The procedures mentioned in each March-sift element will execute sequentially one after another in each state machine’s states to test MUT. After completing the write 0s operations in sift0, the state machine jumps to the next state (sift1) and starts performing read 0s, followed by writing a 1s procedure at each memory location. While performing read operations, the read data will compare with the desired 1s pattern. If the comparison fails, the state machine jumps to the status state and displays the failure information into the status register for the BISR block. The memory tag, fail address, defective cell position, and defect count will be stored in the status register.

The state machine displays the failure information, jumps back to the state from where it arrives, and starts performing the March-sift operation at the next consecutive address location. During testing, if the stop signal triggers, the state machine will jump to the status state and display the same failure information with the pass or fail indication. Similarly, suppose that the halt (halt if an error) is programmed. In that case, the state machine jumps to the status state on any fail condition, displays failure information, and waits for the join signal to continue its operations. In this way, the state machine performs all the procedures mentioned in the algorithm until the last address location, jumps to status state, and displays the testing result. If the fail count is nonzero, the testing result of memory fails; otherwise, it will pass. The completion signal will also be active if the test finishes.

### 2.2. Fault Repair

The memory defects are often increasing by increasing memory size and the high-end technology in recent memory-based devices. These increasing defects are tackled only by the BISR method as it is less expensive, fast, and has a high repairable rate than the other methods. The defective memory part will be replaced with the spare memory, which is not faulty. Therefore, fault repair method by BISR is treated as a reliable and cost-effective solution [21]. The concept of a defective part’s replacement with the redundant block will understand by taking an example of the car Stepney. The car Stepney is a spare tire, and it will be replaced by the defective tire when it gets a puncture. 

This research study proposed the fault repair approach by the built-in redundancy analysis (BIRA) method. The BIRA block consists of redundancy analysis (RA), Fault Table (FT), Buffer, and a redundancy signature register (SR). The faulty addresses are stored in the redundancy logic. The data will be accessed from the redundant locations if those locations are available in RL while memory is operated. The faulty addresses are programmed during memory BIST, and an overflow bit is used to indicate the overflow of the false addresses. This overflow bit indicates that there are more failing addresses than that of the repair cell. The BIRA block is shown in Figure 3.

The redundancy analysis is simple and straightforward, and it selects minimum numbers of spare rows and spare columns to cover the faulty cells. It starts immediately after the BIST passes the status about the fail information. The RL block uses a fault table (FT) to store fault addresses. Based on the fault information, the fault address is stored in the fault table every time if it is not available previously. An overflow bit indicates to the BIST controller that the fault address table is packed, and no other space remains to store the new faulty address.

The repair analysis (RA) is performed on the incorrect addresses available in the fault table. The minimum number of spare rows and spare columns are calculated to access the error-free contents from the spare redundancy in place of the memory’s defective cells. The RA coordinates with BIST to develop the repair strategy as per the available redundancy. The repair information is stored in the signature register (SR), whereas the false information is stored in the fault table. The buffer is used to store the fault addresses and the fault position. It helps when there is more than one faulty bit detected from a particular defective address. It contains the faulty bit address to provide a repair solution. The memory or fault is unrepairable only when FT cannot store the fault address or no redundant cell is available.

The BISR flow is shown in Figure 4, and it consists of the following steps.

The memory BIST block tests the MUT for faults. If fault detects, the BIST generates the failure information, and if no defects, the BIST stops.Once the BIST programs failure information, the repair process starts. The RA block reads the faulty addresses and compares them with the addresses previously available in the fault table. If an address is not previously known, it stores it into the fault table; otherwise, it ignores it.Depending on the erroneous information, the RA performs a repair strategy and calculates fault count in a faulty row and faulty column.The repair signature is prepared based on the repair strategy and will store in the signature register.For whichever is higher, the row-defect count or column-defect count, it will allocate first and repeat until it reaches one fault in a particular row or particular column.If the row fault count equals the column fault count, the spare row will assign.If there is only one faulty cell that remains, the spare row allocates.The repair solution is provided for the memory under test whether the fault is repairable or unrepairable. In the case of multiple memories, the same BISR block will share within the numerous memories by selecting one after another.

The proposed repair algorithm workflow can understand by the example taken for a faulty memory shown in Figure 5. The presented approach uses three rows and two columns to repair the memory. The spare status before the repair process starts (r3, c2) and is shown in Figure 5a. The algorithm will first calculate the row-fault count and the column-fault count and then allocate the spare rows and columns as described in the flow. The spare row and spare column allocation are shown from the example and can be seen in Figure 5b. The memory will repair once it performs all the steps of the BISR flow. The same BISR block will share within the multiple memory by selecting one after another.

## 3. Results and Comparison

The proposed BISR block is implemented on the FPGA platform by using the Xilinx tool and on the ASIC platform, using Synopsys Design Compiler. The register transfer logic (RTL) is written by using Verilog HDL (hardware description language). BISR internally consists of test and repair blocks. The results are for the top-level block consisting of both the blocks (test and repair).

### 3.1. Functional Test

The functional test has carried out on the Xilinx simulator. The proposed BISR is implemented using Verilog HDL. The test benches are written compute different faults and their types. The faults at various locations at different positions are injected, and fault types are calculated for negative testing. Some test scenarios are considered for detecting defects and their types by inserting defective bits at various locations in different positions. The simulation results for finding the faults, such as stuck-at-0, stuck-at-1, NPSFs, transition faults, address decoder faults, coupling faults, Write Destructive Faults (WDFs), Read Destructive Faults (RDFs), and Deceptive Read Destructive Faults (DRDFs), are tabulated in Table 2.

The simulation results of the proposed BISR block are shown in Figure 6. The memory is packed with the 0s at w0 state, as shown in Figure 6a, and reading operations start from the third state and will continue till the seventh state and are shown in Figure 6b,c. Every time-read datum is compared with the desired pattern mentioned in the algorithm, and if the failure occurs, the false information is written into a fail information state. With this failure information, the BIRA prepares a repair strategy and provides a repair solution for the memory under test.

The results of the fault and their type detection are received and compared with the other existing methods. We used eight steps in the algorithm to conduct the test and to enhance the results. The proposed method is a short and efficient fault coverage method and finds more fault types and provides an optimal fault coverage than other existing algorithms. Most of the studies cover only SAF, TF, ADF, and some CFs. Although some faults still may exist in the memory. Therefore, we took a step to cover those faults by the presented method of the March sift algorithm. It covers SAF, TF, ADF, CFs, NPSFs, WDFs, RDFs, and DRDFs from memory under test to improve fault coverage. The fault coverage comparison with the proposed and other existing methods is given in Table 3.

The row and the column count for the detected faulty cell are shown in Figure 7. The defective cells are indicated in the table by the cross (x) indicator. The total fault count in a row and the total fault count in a column are calculated and written in the fault count’s appropriate place for two memories of sizes 8 K and 16 K. It can be seen from the simulation result in Figure 6d,e respectively. The fault count comparison is performed for selecting only the row or the column with a higher fault count. When fault row count and fault column count are equal, the fault row repair will be fixed and assigned with a redundant row. If there is only one fault in a particular row or a particular column, the row repair will select and set for a redundant row. The chosen memory is repaired by the steps mentioned in the flow. The spare row and the spare column are (3, 2) used to fix the memory under test for fault.

### 3.2. Synthesis Process

The synthesis process is completed by using the Xilinx synthesis tool and the Synopsys tool Design Compiler. The top-level module consists of BIST, BIRA, and memory wrappers. The synthesis report for the area, timing, and power are obtained.

#### 3.2.1. FPGA Synthesis

The package layout view of the Xilinx FPGA is shown in Figure 8, and the detailed analysis for the proposed BISR block is received on the FPGA platform as follows.

The Xilinx tool specification is given in Table 4. The area summary is shown in Table 5, where the hardware details, such as the number of slices, flip-flops, and the I/O buffers, are estimated.

The proposed BISR block is compared with the other existing approaches for the area used and is tabulated in Table 6. These approaches are good enough for providing the repair rate, but they have more area penalties when compared with the proposed method. The timing/delay and the maximum frequency outcomes are tabulated in Table 7.

The area overhead for the proposed BISR and the other existing approaches is determined. The area overhead is estimated as per the area report of the proposed block consisting of the slice registers count. The other existing methods for comparison are also implemented on the Xilinx platform, and from the obtained results, the area, the timing, and the area overhead percentage are determined.

#### 3.2.2. ASIC Synthesis

The results have also developed on the ASIC platform for the proposed and other BISR approaches in this research study. The BISR block is synthesized by Design Compiler with the specifications of 32 nm technology, using HVT slow library cells. The obtained result for hardware utilization in terms of cell count is tabulated in Table 8.

The synthesis process for the proposed BISR and the different existing approaches target area, timing, and power. The comparison with other approaches for area, timing, and power are tabulated in Table 9.

The test was conducted on the significant memories of 8 k, 16 k, 24 k, 32 k, 64 k, and 128 k sizes. The area overhead percentage was calculated, and the obtained results are tabulated in Table 10. The graph for the overhead rate is plotted as shown in Figure 9.

The obtained result of the proposed method is compared with the other approaches. Two groups of the memories are formed by combining 8 k and 16 k memory as a Group 1 and 32 k and 64 k memory as Group 2 for the experiment purpose. The comparative results for the proposed and other existing methods for these two groups for area overhead are tabulated in Table 11.

The proposed BISR block with three rows and two-column redundancy is implemented, and experiments are conducted by considering two groups of memories. The Group 1 memory and Group 2 memory areas obtained are 942,542 and 2,004,586 nm^2^, respectively, whereas the areas of the BISR block for these groups of memories are 32,989 and 35,063 nm^2^ received, respectively. The area overhead percentage comparison is tabulated with other methods in Table 12. The overhead percentage by the presented way is obtained about 3.5% and 1.7% for Group 1 and Group 2, respectively. The overhead portion of the BISR block reduces as the size of the memory increases.

The comparative features of the other existing approaches and the proposed method are given in Table 12. We can be concluded from the obtained result that the proposed BISR method is relatively better in terms of the area overhead and the repair rate with the minimum use of the redundancy allocations.

The repair rate is compared with the approaches discussed and the proposed method. It is concluded that, from the results and by the study of other existing BISR methods, the repair rate will increase by increasing the spare row and column count, but it results in the penalty of more area overhead. The estimated chart between repair-rate percentage and the spare-row/column matrix is plotted in the chart of Figure 10, and the overhead portion is given in Table 13.

## 4. Discussion

Memory test and repair is a widely used method for enhancing the yield of SoC-based products. The process consists of two different step’s fault test and fault repair for any memory under test. Many researchers have presented these two steps separately as a new finding. However, we produced this technique both to test and repair the memories for SoC-based devices. The fault test has been covered in the research [22,23,24,25] to test the embedded memory for faults. The stuck-at fault, transition fault, address decoder fault, and coupling faults are computed. The fault test method uses a different March test algorithm to find the defects in the memory. The March c and March c+ algorithms discussed in the research [25] enhance the fault coverage by presenting the March Y algorithm for memory in the study. This March Y algorithm can detect the stuck-at fault, transition fault, address decoder fault, and some coupling faults, whereas the memory test approaches presented [23,24] are not enough to catch all coupling faults. The March SS is given in the research [22] to compute the faults, and it detects SAF, TF, ADF, and some CFs with a comparatively high area overhead. To tackle possibly all types of defects in the memory in this research, we proposed a March-sift algorithm that successfully detects almost all types of faults, such as SAF, TF, ADF, CFs, NPSFs, WDFs, RDFs, and DRDFs, from memory under test. The BIST approaches in the research [10,15] are also presented in the context of self-testing. Comparative results indicate that the proposed approach is better in fault coverage with a minimum area overhead.

As indicated, the proposed BISR scheme in this research targets both test and repair mechanisms. The repair process starts when the test controller provides false information to the BIRA block. The repair strategy is prepared as per the steps described in the repair flow, and the memory is repaired by allocating spare rows and columns. The research studies [15,16,17,26,27] present repairs to the memory blocks by using the calculated redundancy. Chen et al. proposed the BISR scheme using Maximum-size local bitmap (MLB) and FSM [15]. The MLB and level-based buffer (LBB) sizes are reasonably larger than the fault table (FT), and the buffer used in our approach used for the same purpose. The BISR area with the memory found in this method is compared by the proposed method, and the overhead area is found reasonably higher than the proposed BISR method for the same memory size. As presented Reference [26], the Cresta algorithm needs more sub-analyzers to repair the memory. It tests all the faulty cells in the memory, and it requires the row address and column address of all the defective memory cells to provide the repair solution. Therefore, the multiple-bit-failure memory is difficult to repair and needs more area and more redundancies to repair the memory while implementing.

The BISR method of a research study [27] is offered for both test and repair memories; however, the technique lags in fault coverage and repair rate calculation. It proposed a memory test controller based on the March c algorithm and modified it to catch the memory’s faults and types. However, still, some fault types, such as WDFs, RDFs, and DRDFs, are not covered in the research. Additionally, the repair method may improve to increase the repair rate further. The repair rate is further increased in the presented method by the increase of one extra row.

The proposed BISR block in this research study aims to enhance the area overhead, timing, and repair rate. The other existing BISR approaches of the studies [15,16,17,26] are compared with the presented scheme of the BISR module and found with a large spare row–column matrix, the repair rate is reasonable, and the area overhead is reasonably higher. It is noticed that, from the obtained results, the repair rate and area overhead for the proposed BISR scheme is better with the discussed existing approaches with the use of three spare rows and two sparse columns. The area, timing, power, and repair rate are given in the results section. We have noticed from the repair-rate graph that the repair rate increases with the increase of spare row–column.

## 5. Conclusions

The redundancy allocation method is used to repair the memories of the recent SoC-based devices. The spare row and spare column are used when the faults are existing in the memory under test. The BIST block tests the memories for faults and computing faults and their types from the embedded memories. Different types of defects are calculated, and the false information is prepared to provide to the BIRA block for the start of the repair process. The BIRA block repairs the memory by using the redundancy allocations method. The obtained results on the FPGA and ASIC platforms for the area, timing, and repair rate are discussed and compared with other approaches. It is noticed that the proposed method is a good choice to test and repair the memories for recent SoC-based devices. The only limitation is area overhead when integrating with the SoC to develop the current embedded system-based product. The presented BISR scheme can modify the next-generation systems to test and repair the memories of large sizes as the technology enhances.

## Figures and Tables

**Figure 1 micromachines-12-00811-f001:**
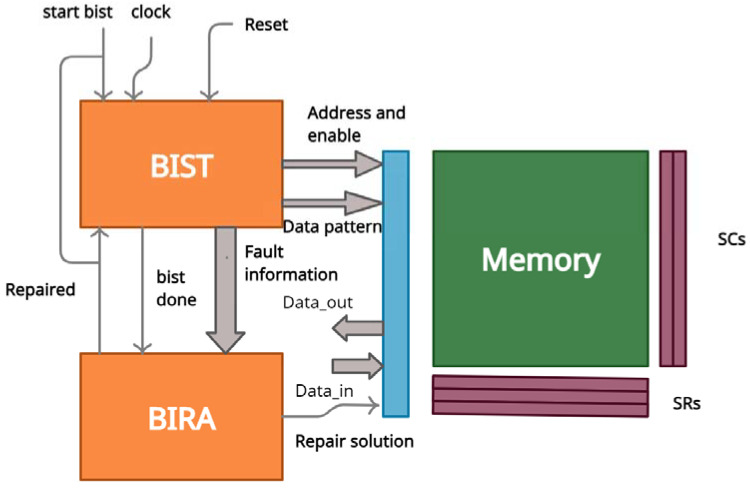
Proposed built-in self-repair (BISR) architecture.

**Figure 2 micromachines-12-00811-f002:**
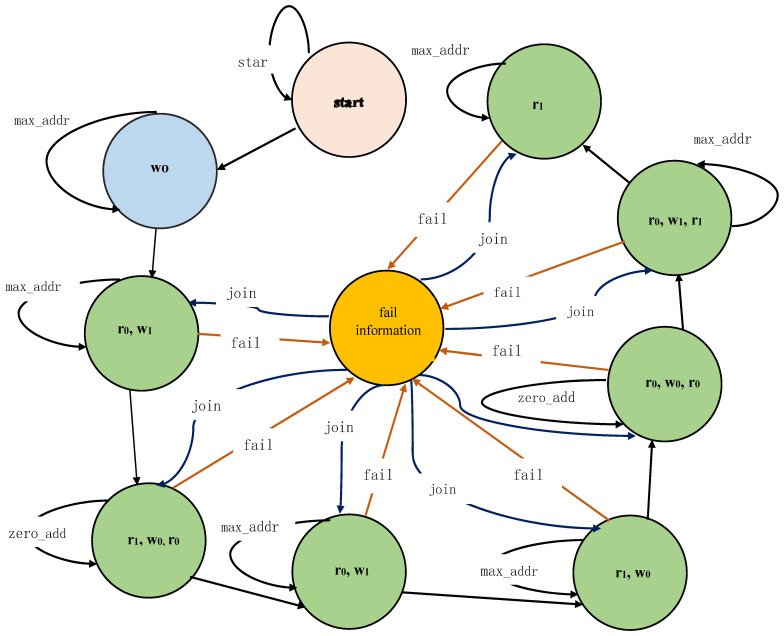
Built-in self-test (BIST) controller state machine.

**Figure 3 micromachines-12-00811-f003:**
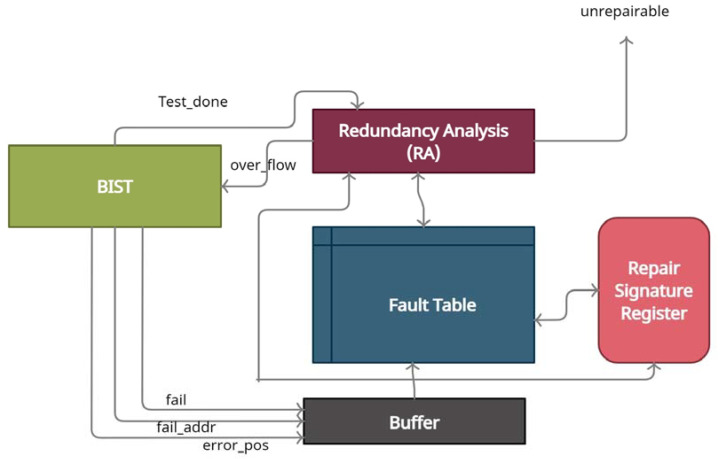
The built-in redundancy analysis (BIRA) block diagram.

**Figure 4 micromachines-12-00811-f004:**
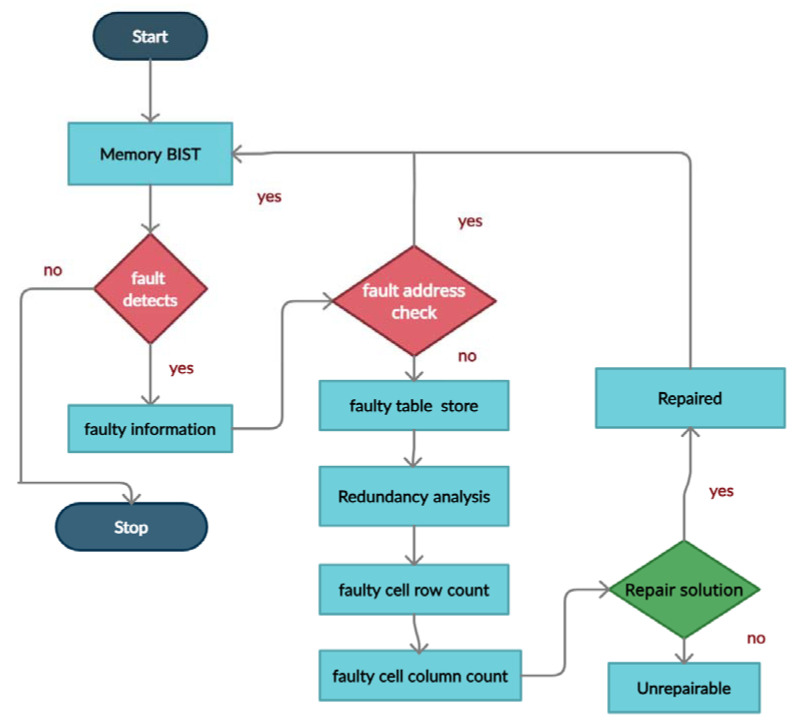
The built-in self-repair (BISR) operational flow.

**Figure 5 micromachines-12-00811-f005:**
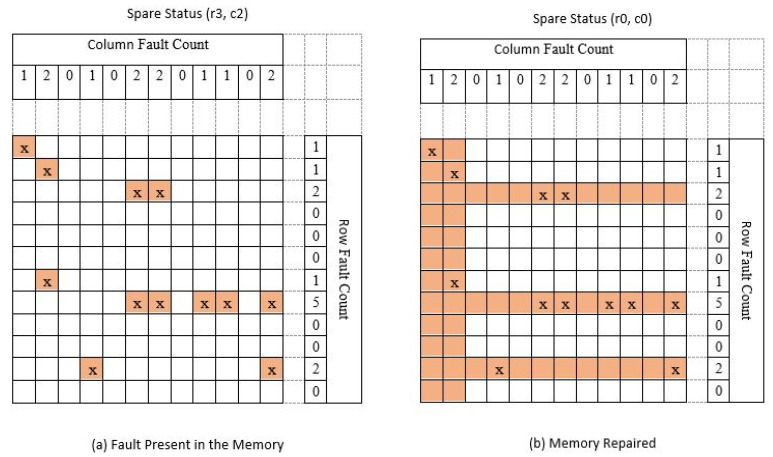
Memory with faults and repair process by the proposed BIRA scheme: (**a**) fault present in the memory under test; (**b**) memory repaired by the proposed method.

**Figure 6 micromachines-12-00811-f006:**
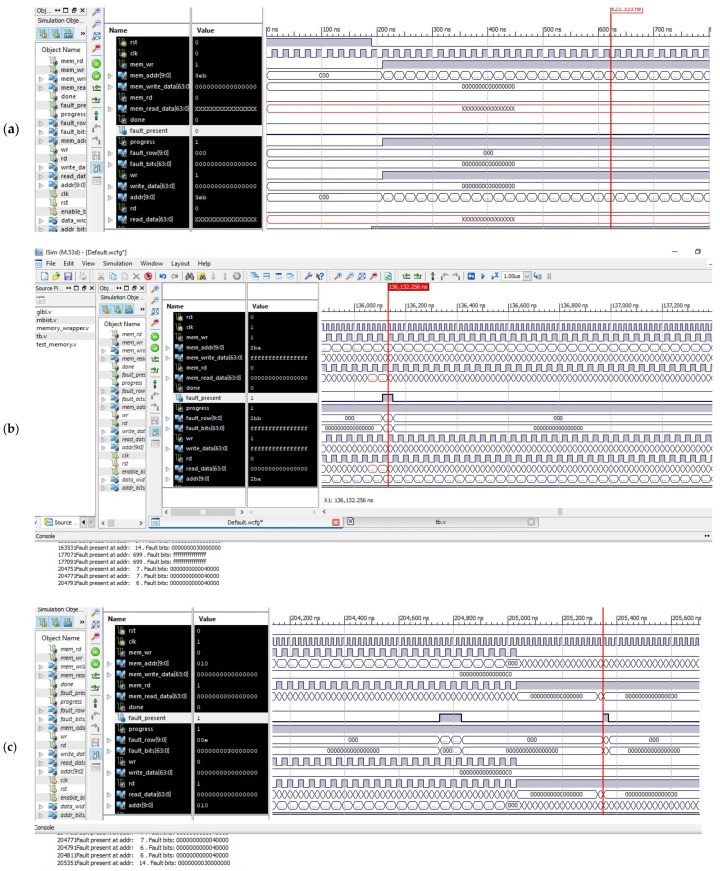
Simulation results for the proposed BISR block: (**a**) writing memory with all 0s; (**b**) BISR operations for performing read–write operations into and from memory; (**c**) BISR operations fault and type detections and store fault information; and (**d**) BISR flow and fault count detection row–column wise and memory repair for 8 k memory. (**e**) BISR flow and fault count detection row–column wise and memory repair for 16 K memory.

**Figure 7 micromachines-12-00811-f007:**
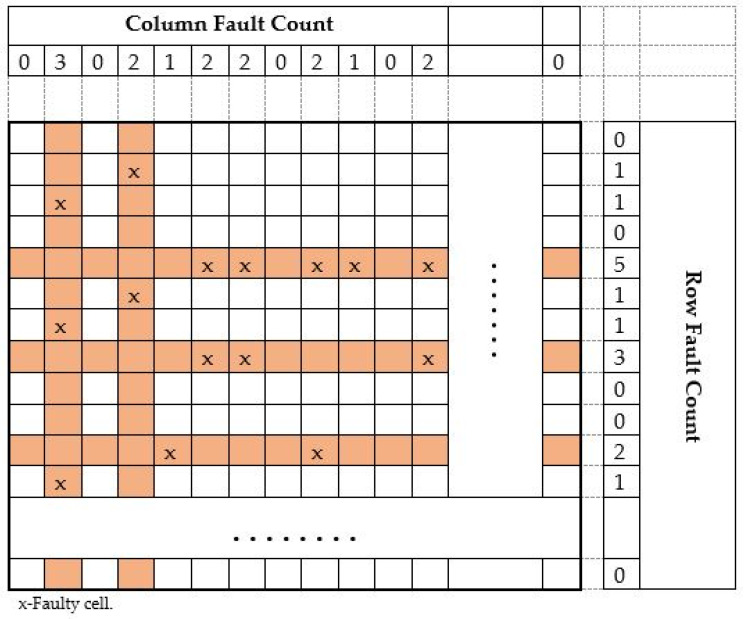
Total fault count row-wise and column-wise 16 Kbytes memory.

**Figure 8 micromachines-12-00811-f008:**
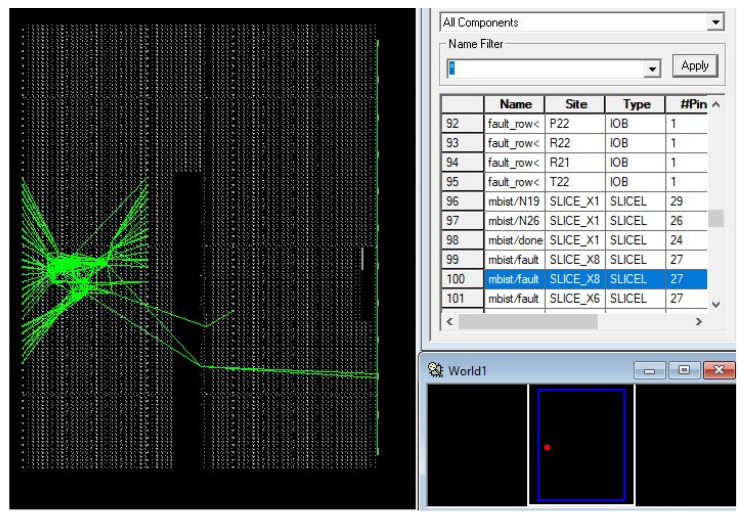
Package layout view.

**Figure 9 micromachines-12-00811-f009:**
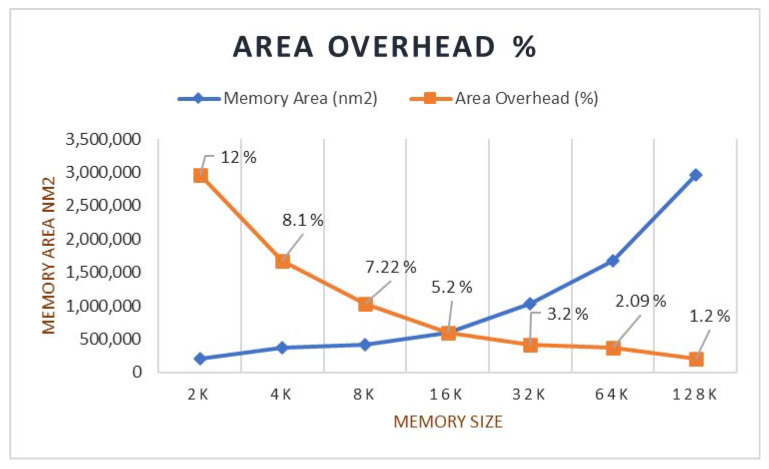
Different memory models versus BISR area overhead.

**Figure 10 micromachines-12-00811-f010:**
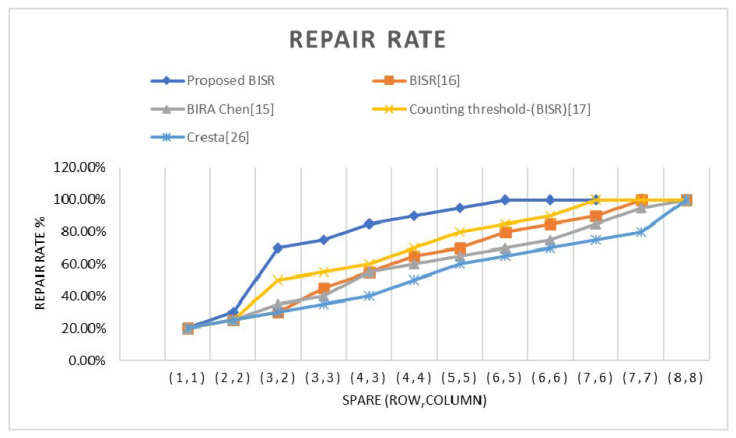
The estimated chart between repair-rate percentage and the spare-row/column matrix.

**Table 1 micromachines-12-00811-t001:** An Algorithm.

March-Sift Algorithm		
act on all memory locations(ml), any address order ascending or descending		sift0: ↕ (w0);
for a := 0 to h do	start from first (0) to last (h) address (a);	
write 0 to ml[a];	write 0 to memory location (ml)	
end;		
act on all memory locations (ml), in sequence ascending order		sift1: ↑ (r0, w1);
for a := 0 to h do	start from first (0) to last (h) address (a)	
read ml[a];	read ml and compare with expected value (0), if mismatch record the result	
write 1 to ml[a];	write 1 to memory location (ml)	
end;		
act on all memory locations (ml), in sequence descending order		sift2: ↓ (r1, w0, r0);
for a := h down to 0 do	start from last (h) to first (0) address (a)	
read ml[a];	read ml and compare with expected value (1), if mismatch record the result	
write 0 to ml[a];	write 0 to memory location (ml)	
read ml[a];	read ml and compare with expected value (0), if mismatch record the result	
end;		
act on all memory locations (ml), in sequence ascending order		sift3: ↑ (r0, w1);
for a := 0 to h do	start from first (0) to last (h) address (a)	
read ml[a];	read ml and compare with expected value (0), if mismatch record the result	
write 1 to ml[a];	write 1 to memory location (ml)	
end;		
act on all memory locations (ml), in sequence ascending order		sift4: ↑ (r_1_, w_0_);
for a := 0 down to h do	start from first (0) to last (h) address (a)	
read ml[a];	read ml and compare with expected value (1), if mismatch record the result	
write 0 to ml[a];	write 0 to memory location (ml)	
read ml[a];	read ml and compare with expected value (0), if mismatch record the result	
end;		
act on all memory locations (ml), in sequence descending order		sift5: ↓ (r_0_, w_0_, r_0_);
for a := h to 0 do	start from last (h) to first (0) address (a);	
read ml[a];	read ml and compare with expected value (0), if mismatch record the result	
write 0 to ml[a];	write 0 to memory location (ml)	
read ml[a];	read ml and compare with expected value (0), if mismatch record the result	
end;		
act on all memory locations (ml), in sequence ascending order		sift6: ↑ (r_0_, w_1_, r_1_);
for a := 0 down to h do	start from first (0) to last (h) address (a)	
read ml[a];	read ml and compare with expected value (0), if mismatch record the result	
write 1 to ml[a];	write 1 to memory location (ml)	
read ml[a];	read ml and compare with expected value (1), if mismatch record the result	
end;		
act on all memory locations (ml), in sequence ascending order		sift7: ↕ (r1);
for a := 0 to h do	start from first (0) to last (h) address (a)	
read ml[a];	read ml and compare with expected value (1), if mismatch record the result	
end;		

**Table 2 micromachines-12-00811-t002:** Faults and their type detection from memory under test.

Faulty Addresses	Read Data with Faulty Bits	Fault-Type Detection
Case I NPSF		
0	0000000000000000	No fault
6	0000000000040000	NPSF at cell 18
7	0000000000040000	NPSF at cell 18
Case II Stuck-at-0 Faults		
8	ffff fffe ffff ffff	Stuck-at-0 fault at cell 32
18	ffff ffff ffff 3fff	Stuck-at-0 fault at cell 12–13
Case III Stuck-at-1 Faults		
22	0000000070000000	Stuck-at-1 fault at cell 28–29
58	0000000000000300	Stuck-at-1 fault at cell 8–9
109	0001000000000000	Stuck-at-1 fault at cell 48
Case IV transition faults and address decoder faults		
120	0000000000000000	No fault
127	0000000000004000	Transition fault at 14
367	ffff ffff fffb ffff	Transition fault at 17
455	0000000001000000	Transition fault at 24
495	null	Address Decoder fault will write to location 511
511	ffff ffff ffff ffff	Address Decoder fault
511	0000000000000000	Overwrite
Case IV coupling faults		
699	0000000000000000	No fault
723	0000000000000000	Coupled with cell 14 at location 1009
786	ffff ffff ffff fff7	Coupling fault at cell 4 with cell 5 at location 723
820	ffff ffff ffff dfff	Cell 14 Coupled with a cell 56 of location 699
Case V Write disturb faults (WDFs)		
887	0000000000000010	Write ‘0’ but Cell 5 become ‘1’
967	ffff ffff ffff efff	Write ‘1’ but Cell 13 become ‘0’
Case VI Read disturb faults (RDFs)		
994	0001000000000000	Read ‘0’ but Cell 49 become ‘1’
1009	ffff ffff 7fff ffff	Read ‘1’ but Cell 32 become ‘0’
Case VII Deceptive read destructive Faults (DRDFs)		
1012	0000000000400000	Read ‘0’ it returns ‘0’ but cell become ‘1’ this appears in the next consecutive read at cell 23.
1019	ffff ffff fffdffff	Read ‘1’ it returns ‘1’ but cell become ‘0’ will appear in next consecutive read.

**Table 3 micromachines-12-00811-t003:** Fault test approaches and comparison.

Test	March Elements	Fault Coverage	Features
March [22]	{c(w0); ⇑ (r0, w1, w0, w1); ⇑ (r1, w0, w1); ⇓ (r1, w0, w1, w0); ⇓ (r0, w1, w0)}	SAF, TF, ADF, some CFs	Complex with Moderate fault coverage
March Test Algorithm [23]	c (w0); ⇑ (r0, w1, w1, r1); (r1, w0, w0, r0); ⇓ (r0, w1, w1, r1); (r1, w0, w0, r0); c (r0);	SAF, TF, ADF, some CFs	Complex with reasonable fault coverage
March C [24]	{c(w0); ⇑ (r0, w1); ⇑ (r1, w0); ⇓ (r0, w1); ⇓ (r1, w0);c(r0)}	SAF, TF, ADF	Simple low fault coverage
March Y [25]	c (w0); ⇑ (r0, w1, r1); ⇑ (r1, w0, r0); ⇓ (r0, w1, r1); ⇓ (r1, w0, r0); c (r0);	SAF, TF, ADF, some CFs	Moderate and moderate fault coverage
Proposed March-sift	{↕ (w_0_); ↑ (r_0_, w_1_); ↓ (r_1_, w_0_, r_0_); ↑ (r_0_, w_1_); ↑ (r1, w0); ↓ (r0, w0, r0); ↑ (r0, w1, r1); ↕ (r1);}	SAF, TF, ADF, NPSF, CFs, WDFs, RDFs, DRDFs	Reasonable with Optimal fault coverage

**Table 4 micromachines-12-00811-t004:** Design Specifications on FPGA.

Tool	Xilinx
Product Version	ISE 12.1
Family	Virtex 6
Target Device	xc6vlx75t-3ff484
Package	FF484
Speed	−3

**Table 5 micromachines-12-00811-t005:** Device-utilization summary (estimated values) for proposed BISR top on Virtex-6.

Logic Utilization	Used	Available	Utilization
Number of Slice Registers	143	93,120	0%
Number of Slice LUTs	253	46,560	0%
Number of fully used LUT-FF pairs	139	220	69%
Number of bonded IOBs	86	240	35%
Number of Block RAM/FIFO	3	156	2%
Number of BUFG/BUFGCTRLs	1	32	3%

**Table 6 micromachines-12-00811-t006:** Comparison of the area with the existing methods and proposed BISR approach.

Slice Logic Utilization	BIRA Chen [15]	BISR [16]	Counting Threshold (BISR) [17]	Proposed Method (BISR)
Number of Slice Registers	343	295	188	143
Number used as Flip Flops	342	295	188	143
Number of Slice LUTs	409	387	287	253
Number of occupied Slices	176	156	70	59
Number of LUT Flip; Flop pairs used	227	201	165	107
Number with an unused; Flip Flop	207	186	73	80
Number of RAM(8 K) (1024 × 64-bit dual-port RAM)	4	4	2	3

**Table 7 micromachines-12-00811-t007:** Timing/delay and maximum frequency comparison.

Timing Comparison					
March Algorithm	Minimum period ns	Minimum input arrival time before clock ns	Maximum output required time after clock ns	Maximum combinational path delay ns	Maximum Frequency MHz
Proposed BISR	3.622	2.218	0.567	1.63	276
BISR [16]	5.32	2.61	4.37	2.23	203
Counting threshold (BISR) [17]	4.641	2.042	4.04	1.97	215.450
Cresta [26]	6.239	4.13	5.396	3.93	160.279

**Table 8 micromachines-12-00811-t008:** ASIC area report of the proposed BISR.

Cell Count	Proposed BISR
Hierarchical Cell Count	5
Hierarchical Port Count	19
Leaf Cell Count	656
Buf/Inv Cell Count	209
Buf Cell Count	9
Inv Cell Count	193
CT Buf/Inv Cell Count	0
Combinational Cell Count	583
Sequential Cell Count	187
Macro Count	0

**Table 9 micromachines-12-00811-t009:** Comparison area, timing, and power for the proposed and other BISR approaches.

Design	Area (Cell)	Frequency (MHz)	Power (µw)
BIRA Chen [15]	2957	250	197.2497 (3.7087 × 10^7^ pW (leakage) and 73uW Dynamic)
BISR [16]	2376	320	147.732 (3.0626 × 10^7^ pW (leakage) and 98uW Dynamic)
Counting threshold (BISR) [17]	2002	320	97.2497 (2.787 × 10^7^ pW (leakage) and 69uW Dynamic)
Proposed Method BISR	1861	350	109.8623 (2.5087 × 10^7^ pW (leakage) and 69uW Dynamic)

**Table 10 micromachines-12-00811-t010:** Various memory models versus BISR area.

Memory Size	Memory Area (nm^2^)	BISR (nm^2^)	Area Overhead (%)
4 k	367,095	29,932	8.1
8 k	423,727	30,597	7.22
16 K	603,137	31,756	5.2
32 k	1,026,895	33,532	3.2
64 k	1,673,452	34,527	2.09
128 k	2,965,987	35,732	1.2

**Table 11 micromachines-12-00811-t011:** Area overhead comparison.

Memory	BIRA Chen [15]	BISR [16]	Counting Threshold (BISR) [17]	Cresta [26]	Proposed Method BISR
Group 1 (8 k + 16 k)	5.3	4.3	3.6	5.7	3.5
Group 2 (32 k + 64 k)	2.5	2.03	1.75	2.7	1.7

**Table 12 micromachines-12-00811-t012:** Comparative features of different BISR approaches.

BISR Approach	Repair Rate	Spare Allocation (Row, Column)	Memory Test Support (BIST)	Speed	Area Overhead
BIRA Chen [15]	High	High(variable)	Yes	Moderate	High
BISR [16]	High	High (3, 3)	Yes	Moderate	Moderate
Counting threshold (BISR) [17]	Low	High (3, 1)	No	Moderate	Moderate
Cresta [26]	High	Very High (3, 3)	No	Low	High
Proposed BISR	High	Low (3, 2)	Yes	High	Low

**Table 13 micromachines-12-00811-t013:** Area overhead comparison and portion.

	**Spare (Row, Column)**	**Area Overhead**
Proposed BISR	(3, 2)	3.5%
BIRA Chen [15]	(3, 3)	5.3%
BISR [16]	(3, 3)	4.3%
Counting threshold (BISR) [17]	(3, 1)	3.6%
Cresta [26]	(3, 3)	5.7%

## Data Availability

Not applicable.

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
