# Peer review of "Optimal Method for Test and Repair Memories Using Redundancy Mechanism for SoC"

_micromachines, 2021, doi:10.3390/mi12070811_

Round 1

Reviewer 1 Report

The author has answered all of my concerns.

Author Response

Thanks for your valuable comments and suggestions on our manuscript. The suggested changes are incorporated in the text of the manuscript to improve the quality of the paper.

Reviewer 2 Report

The manuscript focuses on the introduction of hardware implementation. However, it lacks of the novelty of test algorithm and redundancy analysis algorithm.  Also, the symbol used in page 4 is difficult to read and understand.

Author Response

Thanks for your valuable comments on our manuscript. We tried our best to respond to your comments to improve the quality of our paper. The required changes are incorporated in the text of the manuscript.

The recent SoC-based devices play a more important role as technology enhances day by day. These modern SoC designs are dense with memory, and the users need more promising features from their devices. A smooth functioning memory test algorithm and architecture are required to maintain the product's reputation. In recent SoC-based devices, the embedded memory area is higher than 95 % of the total chip area [1, 2, 3]. The Built-in Self Repair (BISR) is a widespread scheme to enhance the yield of the memory-based product. Therefore, regress testing embedded memories in today's complex SoC-based systems become necessary to retain the products' reputation in the market. Further, the SoC-based product yield is drastically affected by the memories present in the chip. Therefore, a slight change in the technique to improve the test and repair method for the memory will help in increasing the product yield.

We have taken a step in this research to enhance the existing approaches in terms of the repair rate, area overhead, and power consumption with better fault coverage features and fault type detection technique by the proposed BISR block.

The symbols used on page 4 are removed for your convenience, and the appropriate text is incorporated in Table 1.

Reviewer 3 Report

Basically, it's a pretty detailed paper, the quantitive improvement is clear. However, the architectures you compared with(such as Cresta, BIRA Chen, et al) are not well discussed in this paper, so that I can't understand why your proposed idea is better than the others essentially. Maybe you should further highlight why your BISR and March-Sift algorithm cost less energy and lower area overhead in architectural view(which means it's not intuitive at least for me). Also, the handwritten symbol in Table 1 is ambiguous for me, it costs me a lot of time to fully understand the order of all of the contents, so please revise it. The figures' sizes presented in Part 3(Results and Comparison) also should be refined.

Author Response

Thanks for your valuable comments on our manuscript. We tried our best to respond to your comments to improve the quality of the paper. The required changes are incorporated in the text of the manuscript.

The presented study in this research is compared with the existing approaches of the studies [4,19,26]. With a large spare row-column matrix, the repair rate is reasonable, and the area overhead is reasonably higher in these approaches. As presented in the study [19], the Cresta algorithm needs more sub-analyzer to repair the memory. It tests all the faulty cells in the memory, and it requires the row address and column address of all the defective memory cells to provide the repair solution. Therefore, the multiple bit failure memory is difficult to repair and needs more area and more redundancies to repair the memory while implementing. Whereas in the proposed method of BISR, we use only one analyzer. The repair strategies are prepared by the row fault count and column fault count figure without checking each faulty cell row address and column address. The research method presented in [4], Chen et al. proposed the BISR scheme using Maximum-size local bitmap (MLB) and FSM. The MLB and level-based buffer (LBB) sizes are reasonably larger than the fault table (FT), and the buffer used in our approach used for the same purpose. The BISR area with the memory in this method is compared by the proposed method, and the overhead area is found reasonably higher than the proposed BISR method for the same memory size.

The proposed research study consists of two different step's fault test and fault repair for any memory under test. Fault testing is carried out by the proposed March sift algorithm. The fault testing methods are available in the research studies [21,22,23,24] to test the embedded memory for faults. Most of the studies cover only SAF, TF, ADF, and some CFs. Although some faults still may exist in the memory. Therefore, we took a step to cover those faults by the presented method of the March sift algorithm. It covers SAF, TF, ADF, CFs, NPSFs, WDFs, RDFs, and DRDFs from memory under test to improve fault coverage.

The symbols used on page 4 are removed for your convenience, and the appropriate text is incorporated in Table 1. The figures in section 3 (Results and Comparison) are the screenshots of the tool used for the implementation. As per your suggestion, we tried our level best to improve them to the best of their visibility. Would you please zoom in on the screen for clear visibility? I hope you will find it now in acceptable form.

This manuscript is a resubmission of an earlier submission. The following is a list of the peer review reports and author responses from that submission.

Round 1

Reviewer 1 Report

Optimal Method for Test and Repair Memories using Redundancy Mechanism for SoC

The paper presents a memory test and repair scheme as an attractive solution to tackle different faults in memory due to submicron technologies.

The paper presents a still preliminary study; there is only a functional model of memory in the research, there is only one experiment with such a functional model. For example, the study from the same authors suggests an ASIC implementation for some memories sizes. Hence, it is difficult to observe how the method is optimal in the current paper. The bits addressing data 64 bit is 1024 double that is still tiny embedded memory. Usually, memories are in blocks. Hence the proposed system is not prepared for what is claimed in line 8: " embedded memories of enormous size". There is no information about the memories class (only the functional model), which has been tested this approach; authors should explain why it is optimal but with realistic memory models. 

For example, some experiments with randomised faults are detected by the method. The authors know where the fault is and how to repair it; it seems an experiment ad-hoc. For example, the approach deals with faults with March algorithms; it seems that March approximation can not detect all kinds of faults (2).  How the method detects either Write Disturb Faults (WDFs) or Deceptive Read Destructive Faults (DRDFs) or both?

The paper is very similar to the one already accepted (1) by the authors. The paper(1) is not in the references. Such paper has a more advanced study with memories of size 2Kx64-64Kx64 and implementation in 180nm ASIC technology. The authors should explain how the minor changes presented in the present paper respect reference (1); affect and improve fault detection and repair with accurate memory models. Currently, the precise memory models (sizes, technologies) are not in the present paper.

Minor consideration:

The Xilinx ISE12 is from 2010; the evolution of the technology is at a very high speed. Can the approach deal with submicron technologies like 5-7nm or the currently available academia around 22nm? Can the method be used in external memories like HBM2/3, DDR3-5? It seems only for SRAM.

The quality of some images is low, like in figure 2, 5, 8.

The paper contains promising research; however, given the time frame for corrections, my recommendation is a rejection with the possibility of resubmission when the study presents more experiments with realistic models of memories. 

(1) https://www.jstage.jst.go.jp/article/elex/advpub/0/advpub_18.20210092/_pdf/-char/en
(2) https://www.researchgate.net/publication/233960859_Fault_Detection_with_Optimum_March_Test_Algorithm

Author Response

Thanks for your valuable comments on our manuscript. We tried our best to respond to your comments to improve the quality of our paper. The required changes are incorporated in the text of the manuscript.

Reviewer 2 Report

This paper proposes an idea of memory fault recovery mechanism. Though it looks interesting, there are several concerns. Please refer to the below comments.

  1. I was not able to find novelty from the proposed method. The BIST, fault table, BIRA logic, etc. are typically used components and March test is also a common method for fault test (also, March-sift algorithm is similar to the existing methods). The spare row and column based memory repair methods are also commonly used. I was not able to find any new idea or findings from the paper and it just seems to be an implementation of the commonly used algorithm. In Discussion section, the authors stated that this paper proposes test and repair as a combined one. However, the methods and results presented in the paper look just a combination of the test and repair while I couldn't see any synergistic impacts from both. 
  2. The authors provided simulation results with various fault scenarios. However, I'm not sure whether the fault patterns the authors considered in the paper are realistic or not. The authors only considered very simple fault scenarios. In addition, the fault patterns are quite different across the types of the memory cells. For example, SRAMs have access time distribution of Normal distribution. (e)DRAMs have retention time failures. Thus, the fault test and recovery mechanisms must be different depending on what type of the memory cells we use. However, I was not able to find any assumption on the memory cell type. For realistic scenarios of the memory cells, please find VARIUS model (S. R. Sarangi, B. Greskamp, R. Teodorescu, J. Nakano, A. Tiwari and J. Torrellas, "VARIUS: A Model of Process Variation and Resulting Timing Errors for Microarchitects," in IEEE Transactions on Semiconductor Manufacturing, vol. 21, no. 1, pp. 3-13, Feb. 2008) or models presented in A. Agarwal, B. C. Paul, S. Mukhopadhyay and K. Roy, "Process variation in embedded memories: failure analysis and variation aware architecture," in IEEE Journal of Solid-State Circuits, vol. 40, no. 9, pp. 1804-1814, Sept. 2005. By using these models, one can model the memory faults in a more realistic way.
  3. The authors presented the BISR logic with FPGA. However, typically the BISR logic is implemented in ASIC because FPGA is worse in terms of power, performance, and area. For realistic comparison, the ASIC implementation (at least logic synthesis and/or layout) should be provided for better latency and area comparison. 
  4. The authors should put a huge effort on presentation of the paper. There are so many grammatical errors (particularly for passive and active voice) and figure resolution should be improved. In the below, some of the grammatical errors are shown but, I found a lot more errors when I read the paper. 

Grammatical errors:

 - page 5, line 149: "will record" should be "will be recorded". There are many sentences contains these words. They should also be revised.

 - page 6, line 174: "will compare" should be "will be compared".

 - page 6, line 183: "will store" should be "will be stored".

 - page 6, line 196: "by increase" should be "by increasing". 

 - page 6, line 199: "will replace with" should be "will be replaced with".

Author Response

(The authors gave the same response as above.)

Reviewer 3 Report

This paper proposes a spare row and column based built-in self-repair (BISR) method. The method is proposed to receive the optimal repair rate with a low area overhead at the tradeoff for a built-in redundancy analysis (BIRA). March-shift algorithm and built-in self-test (BIST) controller is proposed for BIST implementation. I expect that this paper will be of interest to readers by a minor revision.

- Minor spell check such as case-sensitive (e.g. bisr -> BISR, line 198, page 6) is required.

- The figures and tables need to be revised to increase the power of delivery.

Author Response

(The authors gave the same response as above.)

Round 2

Reviewer 1 Report

Dear Authors, 

Thank you for your answers and the great effort for the resubmission with the tight deadline. My answers to your previous answers go below. And following the comments to your upgrades in red to the text.

Point 1: The 8K, 16K memory size is too tiny, and the method is limited to small memories; the authors do not show how it performs in more significant memories. The method has a substantial limitation. However, we can accept the method will perform similar independently of the size.

Point 2: Taking one of your added methods to detect faults:

“Case VII: The DRDFs appear when a read operation causes the inversion of the cell value and returns the original value. When the data pattern 0s read from the memory location 1012, the pattern 0s will return, but the cell 23 value will change to ‘1’ due to DRDF and will identify in the next read operation.”

How does the system know that it has to read twice in the same address? The method writes “000” and reads “000”, which is the correct results, but a “1” is inferred; how does it differ from the regular operation? As you do not know, the method will have to read all the addresses of the memory minimum twice with an execution time overhead, which is not presented in the paper. The experiment seems built ad-hoc for detecting the faults. There is no fault randomised in the memory and accurate coverage of the defects. Line 109 of the text claims 100% coverage is very ambitious. 

The added fault detect methods (point 1, i.e. DRDF) should modify table 6 and 7 (FPGA synthesis) since adding new methods should increase the size of the finite state machine. 

Point 2 is still very fragile, and the claim is substantial that the method presented performs the best, covering all the faults with the smaller FPGA size, higher work frequency.

Point 3: Here, I see enough differentiation with the previous paper. Thanks for the response. My recommendation is that you explain the differences in the manuscript with reference 29.

Point 4: The part of the synthesis, if the clock frequency in the FPGA is around 280Mhz(180nm ), the ASIC(32nm) should work minimum at GHz. Usually, the freq in FPGA, similar system in an ASIC with the same technology is a factor 10; if you use better technology 180nm to 32nm, the factor can be larger. My recommendation is to revise the outputs from the synthesis tool.

There are no differences in table 11 in the memory area? The extra cells used for some methods do not make enlarge the area. Your method is based on extra memory rows and columns that other methods do not use. It seems that always your method is the best, but there is a trade-off in terms of area overhead.

Point 5: The images have improved in quality; however, Figures 6 and subfigures have only information in the left part with the Name and values. The rest is “00”, “XX”, or impossible to visualise.

My recommendation is still a rejection with the possibility of resubmission when the authors have enough time to improve the paper.